# Weak localized electrons enhance electronic coherence for efficient photocatalytic uranium removal from nuclear wastewater

Yachao Xu[1], Ruolan Zhao[2,3], Youxing Liu[1,2] ✉, Ying Wang[3], Zheng Lin[1], Zongqiang Sun[1], Peng Yu[3], Mingchuan Luo[1] & Shaojun Guo[1] ✉

Photocatalytic uranium removal from nuclear wastewater is a promising strategy for radionuclide mitigation, yet its practical deployment is greatly hampered by inefficient charge separation of electron-hole pairs in photocatalytic systems, which critically limits the generation of reactive oxygen species essential for uranium precipitation. Herein, we propose a approach that leverages weakly localized electrons to enhance electronic coherence in fluorine-functionalized covalent organic frameworks, thereby strengthening charge separation and promoting directional electron transport for greatly boosting photocatalytic uranium removal from nuclear wastewater. We demonstrate that the partial fluorination in covalent organic frameworks induce weak electron localization and strong electronic coherence, which delivers a record-high solar-to-chemicals conversion efficiency of 1.52% and achieves 100% uranium removal efficiency within the pH range of 3-6, out-performing non-fluorinated (0.31%) and over-fluorinated (0.85%) counter-parts. More importantly, a self-designed flow-type reactor achieves 99% uranium removal efficiency and superior U processing capacity of 281.3 g m$^{-2}$ day$^{-1}$ under natural sunlight, significantly surpassing reported photocatalytic systems and meeting World Health Organization discharge limits. Mechanism investigation reveal that the partial fluorination enhanced electronic coherence improves photogenerated carrier transport and reparation for accelerating reactive oxygen species synthesis, thereby promoting uranium removal from nuclear wastewater.

Nuclear power is regarded as a promising new energy to substitute fossil fuels[1,2]. Uranium is the most principal nuclear fuel to trigger the nuclear fission reaction in the process of nuclear power utilization[3,4]. However, the nuclear wastewater is ineluctably generated during the operation of nuclear power plants and the mining process of nuclear mines, which destroy soil, water sources and ecosystems[5,6]. The tradition physical adsorption method for uranium removal from nuclear wastewater suffers from weak adsorption interaction between uranium ions and adsorbents, leading to an unsatisfactory removal efficiency of uranium ion[7,8]. Although chemical adsorption method can efficiently adsorb uranium ions from nuclear wastewater[9,10], difficult post desorption of uranium ion usually results in low repetition rates of adsorbents[11,12].

[1]School of Materials Science and Engineering, Peking University, Beijing, China. [2]Beijing Key Laboratory of Electrochemical Process and Technology for Materials, Beijing University of Chemical Technology, Beijing, China. [3]School of Physics and Electronic Engineering, Harbin Normal University, Harbin, China. ✉e-mail: liuyxpost@pku.edu.cn; guosj@pku.edu.cn

Photocatalytic uranium removal technology, by generating $H_2O_2$ or $\cdot O_2^-$ to react with $UO_2^{2+}$ for forming uranyl superoxide ($UO_2(O_2)$ $\cdot xH_2O$) precipitates[13,14], exhibits prospects for uranium ion removal from nuclear wastewater due to its low cost[15,16], high removal efficiency and no post-treatment desorption[17,18]. Over the past decades, various photocatalytic materials, such as metal oxides, metal sulfides and organic semiconductors, have been explored for uranium extraction from nuclear wastewater[19,20], which primarily focus on generating reactive oxygen species (ROS) to convert $UO_2^{2+}$ into stable uranyl superoxide precipitates[21,22]. However, metal oxide and sulfide-based photocatalysts often suffer from weak light absorption in the visible spectrum, poor stability under harsh wastewater conditions and complex synthetic processes[23,24]. Covalent organic frameworks (COFs), are considered as efficient photocatalysts for uranium removal because its unique donor-acceptor (D-A) structure provides a new platform for the photosynthesis of ROS, including $H_2O_2$, $\cdot O_2^-$, and $\cdot OH$, via oxygen reduction reaction (ORR) and water oxidation reaction (WOR) process for the photosynthesis of $UO_2(O_2) \cdot xH_2O$ precipitates from nuclear wastewater[25,26]. However, adverse photogenerated carrier recombination usually occurs in COFs during photocatalytic uranium removal processes, leading to low solar-to-chemical conversion and unsatisfactory uranium removal efficiency[27,28].

Herein, we report a strategy that leverages the manipulation of local electron distribution to enhance electronic coherence, for the purpose of strengthening directional carrier transport and substantially improving photogenerated electron-hole separation, which boosts photocatalytic uranium removal from nuclear wastewater. The fluorine atom with strong electronegativity was introduced as an electron-regulating motif to modulate the local electron distribution in COF molecules and locally align energy levels, which reinforces electronically coherent transport along the preferred pathways. Our systematic investigations reveal that the TAPT-TPA-2F COF, featuring weak electron localization characteristics, endows enhanced electronic coherence. This electronic modulation effect not only facilitates intraframework directional electron transfer from the donor to acceptor moieties but also effectively mitigates electron-hole pair recombination. The as-made TAPT-TPA-2F COFs deliver a high solar-to-chemical conversion efficiency of 1.52% and achieve 100% uranium removal efficiency, far exceeding those of reported photocatalytic materials. Notably, the TAPT-TPA-2F-based flow reactor achieves a benchmark processing U efficiency of 281.3 g m$^{-2}$ day$^{-1}$ under natural sunlight, surpassing state-of-the-art systems and fulfilling World Health Organization (WHO) safety criteria, highlighting its strong application potential for nuclear wastewater treatment.

## Results

### Design strategy of F regulated local electron distribution in COFs

Photoexcited electron transport from electron donor unit to electron acceptor unit is desirable during the photocatalytic process. Accurately manipulating directional electron transfer through regulating electron localization can hinder the recombination of electron-hole pairs (Fig. 1a–b). To validate this concept, we investigate the regulatory role of electron localization in tailoring electronic coherence within COFs, which in turn enhances directional charge transport. 1,3,5-tris-(4-aminophenyl) triazine (TAPT) and terephthalaldehyde (TPA) were selected as molecular modules for constructing TAPT-TPA COFs. Molecular simulation results reveal that the electron donor and acceptor moieties are spatially segregated within the TAPT and TPA building blocks of the COFs. This spatial segregation in turn establishes intrinsically electronically coherent directional donor–acceptor (D–A) transport pathways, which facilitate the directional migration of photogenerated electrons from the TAPT (donor) to TPA (acceptor) moieties. Furthermore, fluorine atoms with strong electronegativity are employed to modulate the locally asymmetric electronic density

distribution within the acceptor moieties (Fig. 1c–e). This structural engineering strategy preserves the frontier-orbital overlap between donor and acceptor units and meantime maintaining phase matching along the dominant coherent charge-transport pathway, thereby promoting constructive interference that further enhancing directional charge separation and transport. Partial fluorination in TAPT-TPA-2F COFs induce asymmetric electron localization and reinforce electron coherent for promoting directionally electron transfer (Fig. 1f–h). In contrast, relative to the TAPT-TPA-2F COFs, excessive fluorination in the TAPT-TPA-4F COF induces isotropic electronic over-localization, which generates multiple phase-mismatched charge transport pathways and ultimately impairs the efficiency of coherent charge transport (Fig. 1b). Consequently, oxygen molecules adsorbed on the electron acceptor unit are reduced to $H_2O_2$ or $\cdot O_2^-$, which subsequently react with enriched $UO_2^{2+}$ to form stable $UO_2(O_2) \cdot xH_2O$ precipitate (Supplementary Fig. 1)[29].

### Molecular structure and chemical bond characterization

The powder X-ray diffraction (PXRD) techniques were carried out to investigate the crystal structure of different COFs. The peaks at $2\theta = 2.7, 4.9, 5.6, 7.3$ and $26°$ in the XRD pattern of TAPT-TPA-2F COFs correspond to their (100), (110), (200), (210) and (001) crystal facets (Supplementary Figs. 2–4), consistent with simulated A-A stacking structure (Supplementary Figs. 5–7). High-resolution TEM (HRTEM) images of TAPT-TPA, TAPT-TPA-2F and TAPT-TPA-4F reveal well-defined periodic porosity with an apparent pore size of ~3.0 nm, and the resolved lattice fringes can be indexed to the (100) plane (Supplementary Fig. 8). Solid-state 13 C NMR spectra of TAPT-TPA-2F COFs show that the chemical shift, located at around 150 ppm, corresponds to aromatic carbon atoms bonded to fluorine atoms (C-F bonds), indicating successful incorporation of fluorine into TAPT-TPA-2F COFs and TAPT-TPA-4F COFs. The chemical shift, located at 168 ppm, is attributed to C = N imine bonds, indicating the formation of C = N imine bond connected COFs (Supplementary Figs. 9–11)[30]. Fourier-transform infrared (FT-IR) spectra of TAPT-TPA-2F COFs show the vibrational peak, located at around 689 cm$^{-1}$, corresponds to C-F chemical bond (Supplementary Figs. 12–15). Raman spectra of TAPT-TPA-2F COFs show a strong peak at 1620 cm$^{-1}$, ascribed to the imine bond (C = N), indicating the synthesis of imine bond connected COFs (Supplementary Fig. 16)[31]. $N_2$ adsorption–desorption analysis verify that the incorporation of fluorine atoms does not compromise the porous architecture of COFs (Supplementary Figs. 17–18).

### Enhanced electronic coherence and exciton dissociation facilitated by weak electron localization

Ultrafast transient absorption (TA) spectra reveal that TAPT-TPA-2F COFs exhibit a significantly prolonged carrier lifetime ($\tau_{TA} = 35.16$ ps) relative to TAPT-TPA-4F COFs (17.63 ps) and TAPT-TPA COFs (6.32 ps), demonstrating that they exhibit the higher spatial electron-hole separation efficiency (Fig. 2a-2c, Supplementary Figs. 19–24). Both TAPT-TPA-2F and TAPT-TPA-4F COFs exhibit the pronounced built-in electric fields under illumination (Fig. 2d–2f). TAPT-TPA-2F COFs attains the contact potential difference (CPD) plateau more rapidly than TAPT-TPA-4F COFs (Fig. 2g–2i), which is indicative of a faster photoresponse (Supplementary Figs. 25–27). This result further confirms the intrinsically superior charge separation efficiency and carrier mobility of TAPT-TPA-2F COFs[32]. Furthermore, the temperature-dependent photoluminescence measurements demonstrate that TAPT-TPA-2F COFs exhibit the lowest exciton binding energy ($E_e = 43.82$ meV), which is markedly lower than those of TAPT-TPA-4F COFs (55.17 meV) and TAPT-TPA COFs (104.64 meV). These findings collectively confirm that enhanced electronic coherence promotes efficient exciton dissociation (Supplementary Figs. 28–30).

Positron annihilation lifetime spectroscopy reveal that TAPT-TPA-2F COFs have a shorter positron lifetime than TAPT-TPA-4F COFs,

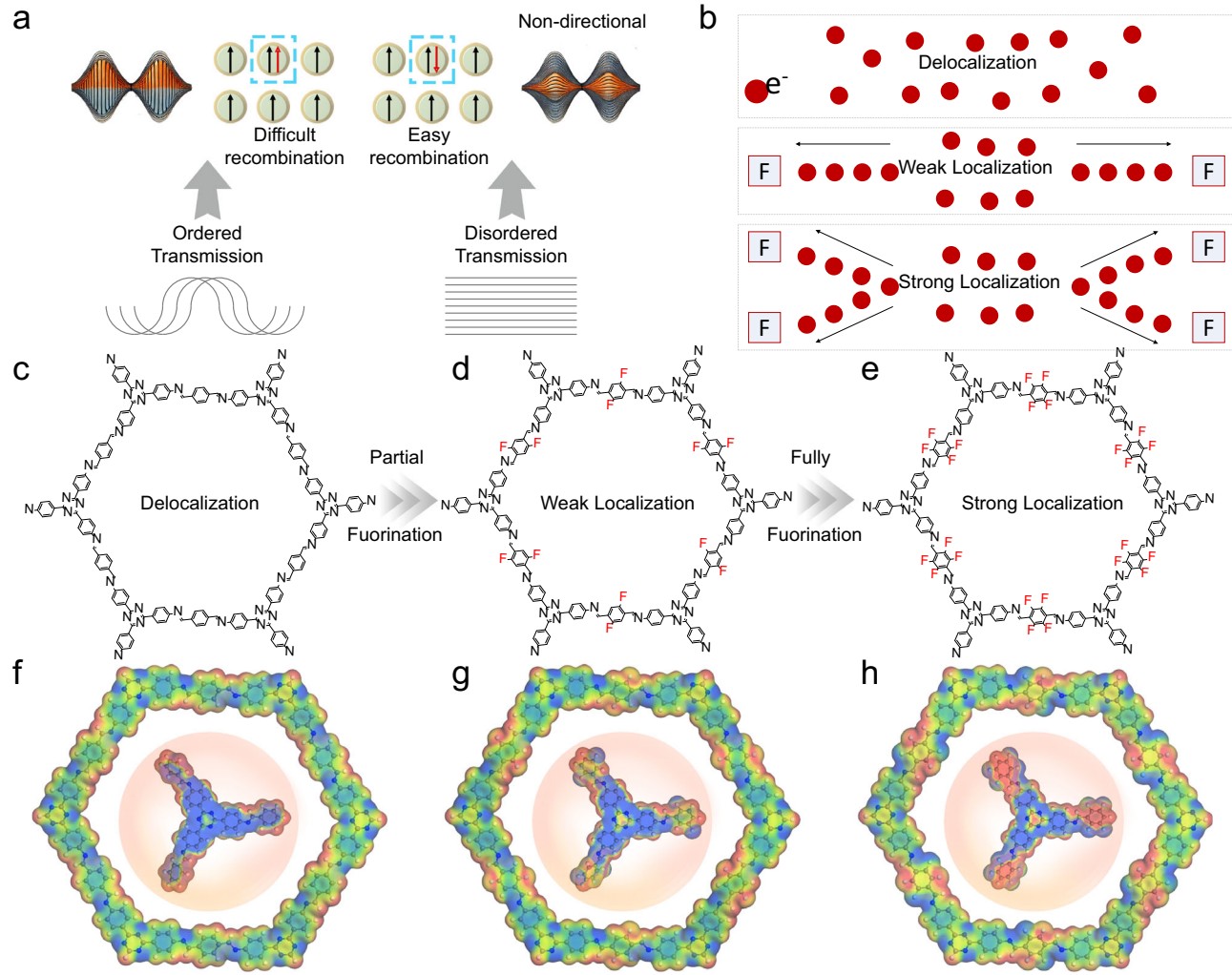

**Fig. 1 | Molecular design of weak-electron-localization. a** Schematic diagram of enhanced electronic coherence promoting directional electron transfer. **b** Electronic directional transmission mechanism. **c** Molecular structure of TAPT-TPA COFs, (**d**) TAPT-TPA-2F COFs and (**e**) TAPT-TPA-4F COFs. Surface electrostatic potentials of (**f**) TAPT-TPA-2F COFs, (**g**) TAPT-TPA-4F COFs and **h** TAPT-TPA COFs.

indicating its higher electron density (Supplementary Fig. 31, Supplementary Table 1). This electron-rich environment is favorable for enhancing photocatalytic performance by effectively facilitating photogenerated electron separation and transfer. Transient fluorescence spectra of TAPT-TPA-2F COFs show that their decay lifetime is $t_1 = 2.1$ ns and $t_2 = 203.5$ ns, significantly higher than those of TAPT-TPA COFs ($t_1 = 1.3$ and $t_2 = 39.9$) and TAPT-TPA-4F COFs ($t_1 = 1.4$ and $t_2 = 68.9$) (Supplementary Fig. 32 and Table 2), further suggesting that partial fluorination–induced directional charge delocalization and coherent charge-transport channels in TAPT-TPA-2F COFs facilitate more efficient photogenerated electron–hole pair separation. Meanwhile, TAPT-TPA-2F COFs also exhibit superior photoelectric conversion performance relative to TAPT-TPA-4F COFs and TAPT-TPA COFs, indicating its excellent electron transport capability (Supplementary Figs. 33–35).

### Photocatalytic uranium removal

To assess the photocatalytic uranium removal performance, all COFs were used for photocatalytic uranium removal at optimal condition of pH=4 (Supplementary Figs. 36 and 37). The as-made TAPT-TPA-2F COFs exhibit a 100% uranium removal efficiency and high uranium removal rate of 0.76% min$^{-1}$, which is higher than those of TAPT-TPA COFs (0.45% min$^{-1}$), TAPT-TPA-4F COFs (0.5% min$^{-1}$) (Supplementary Fig. 38) and reported photocatalytic materials (Supplementary

Table 3). Time-dependent uranium deposition curve shows that the precipitated amount of uranium is increased with light illumination duration, confirming the contribution of light radiation for U(VI) reduction (Supplementary Fig. 39). In addition, we demonstrate that TAPT-TPA-2F COFs also exhibits higher photocatalytic uranium efficiency than TAPT-TPA COFs and TAPT-TPA-4F COFs under natural water (pH = 6.1), neutral (pH = 7) and alkaline (pH = 8.2) condition (Supplementary Figs. 40–42). The as-made TAPT-TPA-2F COFs exhibit a high solar-to-chemical conversion (SCC) efficiency of 1.52%, which is 4.8 and 1.8 times higher than TAPT-TPA COFs (0.31%) and TAPT-TPA-4F COFs (0.85%) respectively, as well as significantly higher than those reported photocatalytic materials (Fig. 3a, Supplementary Fig. 43 and Supplementary Table 4). The apparent quantum efficiency (AQE) profile shows that the maximum AQE of TAPT-TPA-2F COFs is 19.7% under 380 nm-wavelength light radiation (Supplementary Fig. 44). Besides, we find that the introduction of F atom did not significantly alter the energy level structure of three COFs, indicating that the enhancement of photocatalytic performance is contributed from electronic coherence effect (Supplementary Figs. 45–48).

We find that the adsorption kinetic of $UO_2^{2+}$ over all samples follows the pseudo-first-order kinetic model (Supplementary Figs. 49–51, Supplementary Table 5). The uranium adsorption over TAPT-TPA-2F COFs follows Langmuir monolayer adsorption model (Supplementary Figs. 52–55, Supplementary Table 6), which is conductive to the

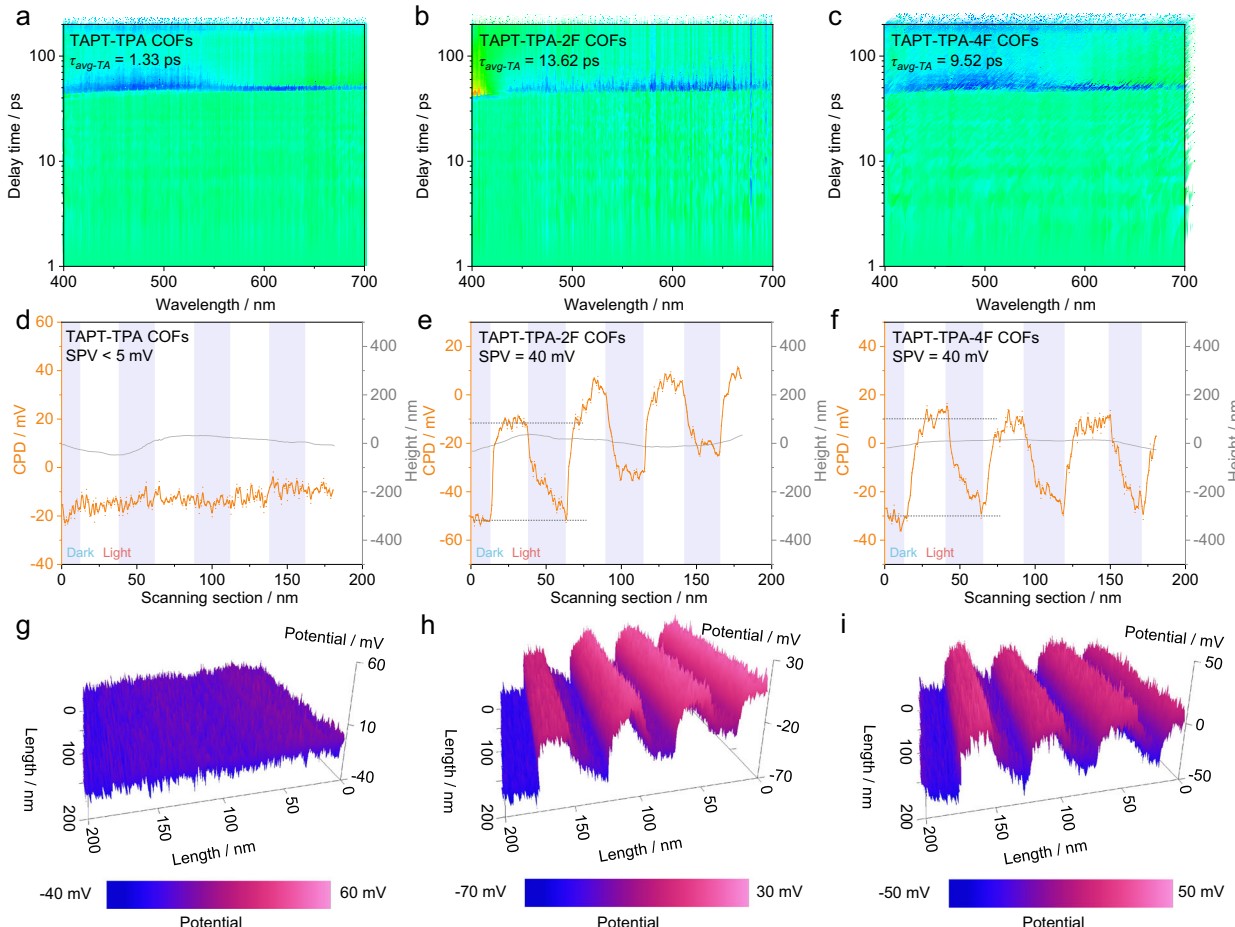

**Fig. 2 | Weak localization-enhanced electronic coherence promotes electron-hole separation.** a–c 2D mapping TA spectra and (d–f) surface photovoltage response of TAPT-TPA-4F COFs measured by Kelvin probe force microscopy under alternating dark and light conditions. The CPD (orange line) and corresponding surface height profile (gray line) were recorded simultaneously to correlate photovoltage behavior with surface morphology. g–i In situ AFM-derived surface potential distribution of COFs under external stimulation, showing a gradual potential variation from −50 mV to +50 mV across a 200 nm × 200 nm region. TAPT-TPA (**a**, **d**, **g**), TAPT-TPA-2F (**b**, **e**, **h**) and TAPT-TPA-4F (**c**, **f**, **i**) COFs.

reaction of $UO_2^{2+}$ and ROS to form uranyl superoxide complexes (Supplementary Figs. 56–57)[33]. To evaluate the feasibility of large-scale and continuous uranium removal, we constructed a flow-type TAPT-TPA-2F COFs-based photocatalytic system (Fig. 3b). This system achieves a high uranium removal efficiency of 99% even under the ultra-low $UO_2^{2+}$ concentrations (1ppm) (Fig. 3c), which is below the WHO discharge limit of 30 ppb. More importantly, it delivers a benchmark nuclear wastewater processing capacity of 281.3 g U m$^{-2}$ day$^{-1}$, significantly surpassing previously reported photocatalytic systems and demonstrating strong potential for scalable application. Furthermore, cycling tests confirm the excellent reusability and operational stability of TAPT-TPA-2F COFs over multiple runs (Fig. 3d). PXRD profiles of TAPT-TPA-2F COFs used for uranium removal reveals integrity of porous structure, which confirm that the crystal structure and chemical bond of TAPT-TPA-2F COFs were not disrupted during the uranium removal process (Supplementary Fig. 58).

**Photocatalytic mechanism**

To reveal the essential relationship between electronic coherence and photocatalytic uranium removal performance, both experimental and DFT calculations were carried out. Chemical probe transfer experimental results show that the concentration of $H_2O_2$ and $\cdot O_2^-$ over TAPT-TPA-2F COFs is 492.5 μmol L$^{-1}$ and 1501.1 μmol L$^{-1}$, respectively, significantly higher than those of TAPT-TPA COFs (The concentration of $H_2O_2$ and $\cdot O_2^-$ is 148.1 mol L$^{-1}$ and 210.8 mol L$^{-1}$,

respectively) and TAPT-TPA-4F COFs (The concentration of $H_2O_2$ and $\cdot O_2^-$ is 365.6 mol L$^{-1}$ and 730.4 mol L$^{-1}$, respectively) (Supplementary Figs. 59 and 60), which is consistent with the electron paramagnetic resonance (EPR) results (Supplementary Fig. 61). The molecular dynamics simulation results reveal that the introduction of F atom can significantly promote the adsorption of $UO_2^{2+}$ over TAPT-TPA-2F COFs and TAPT-TPA-4F COFs, which accelerates the reaction of $UO_2^{2+}$ and ROS for the formation of uranyl superoxide (Supplementary Figs. 62 and 63).

DFT calculation reveals that the strong electronegativity of F atom regulates the electron localization distribution (Supplementary Figs. 64–66). TAPT-TPA COFs exhibit a completely delocalized electron distribution, accompanied by the disorderly electron transfer, resulting in severe photogenerated carrier recombination. For TAPT-TPA-2F COFs, the F atom with strong electronegativity induces the asymmetric electron cloud distribution, facilitating the direct photogenerated electron transport from bonding to anti-bonding orbitals during photocatalysis[34]. In addition, TAPT-TPA-2F COFs exhibit a lower p-band center (1.47 eV) and higher density of states near the Fermi level relative to TAPT-TPA-4F COFs (p-band center of 1.81 eV) (Fig. 4a–b), which can decrease the overlap between the C 2p and F 2p orbitals (10.10%) in TAPT-TPA-2F COFs relative to TAPT-TPA-4F COFs (The overlap between the C 2p and F 2p orbitals is 20.97%). These results can prevent the excessive electron localization for enhancing the electron transfer

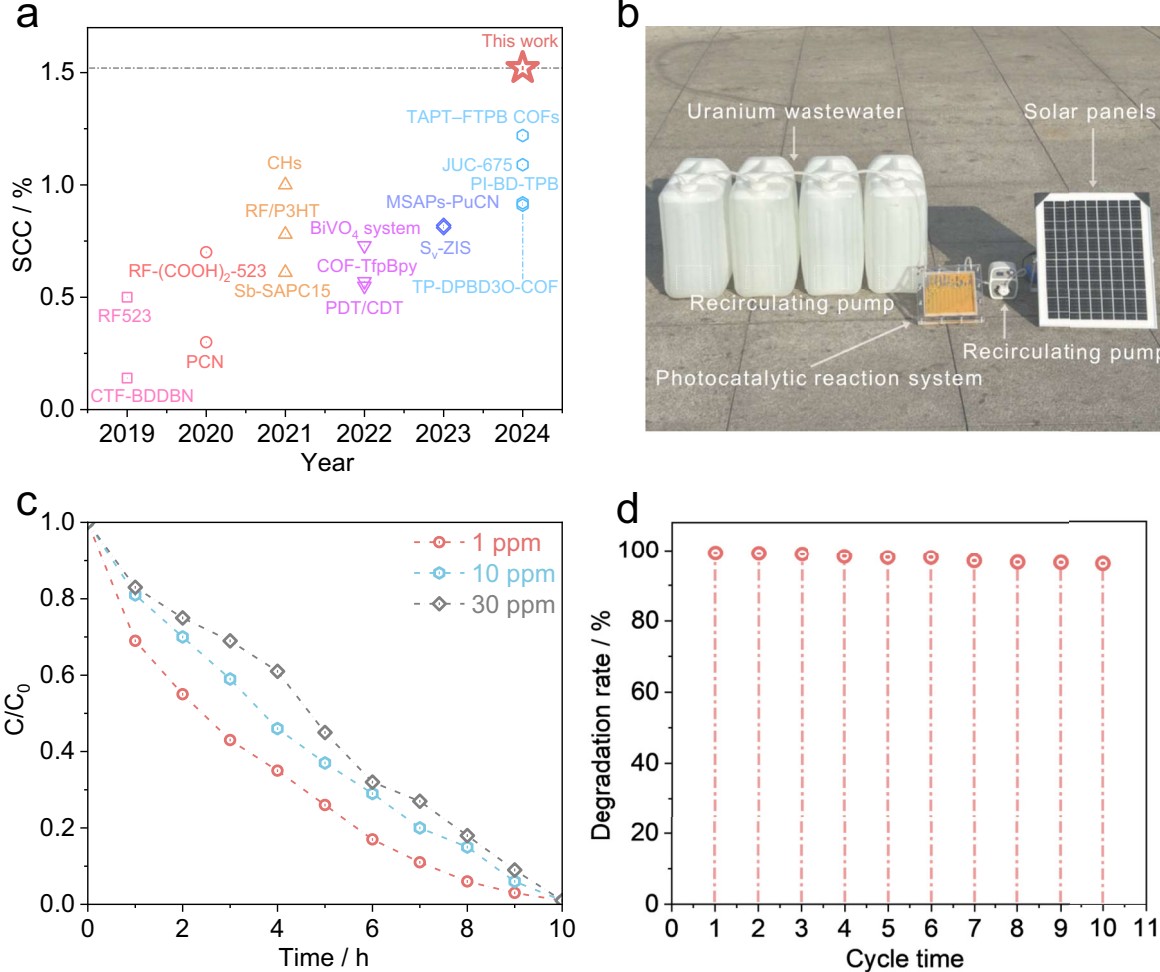

**Fig. 3 | Weak electron localization enhances electronic coherence for efficient uranium removal. a** SCC comparison between TAPT-TPA-2F COFs and state-of-the-art photocatalysts (Supplementary Table 4). **b** A self-developed continuous photocatalytic reaction system for uranium removal. **c** Removal efficiencies of U(VI) using TAPT-TPA-2F COFs under different uranium concentrations (100 L of simulated nuclear wastewater). **d** The cyclic stability of a self-developed continuous photocatalytic uranium removal reaction device.

(Fig. 4c–d)[35]. Thus, TAPT-TPA-2F COFs with weak electron localization exhibit excellent electron-hole separation efficiency for enhancing photocatalytic uranium removal. The fluorination-induced local energy level engineering not only guides electron transfer pathways toward acceptor moieties but also strengthens electronic coherence, which in turn boosts directional electron transport and improves electron–hole separation dynamics. For TAPT-TPA-2F COFs, the partial fluorination-triggered weakening of electron localization contributes to enhanced electronic coherence, alongside favorable modulation of directional electron mobility and band alignment[36]. In contrast, excessive localization in TAPT-TPA-4F COFs decrease the energy band density near the Fermi level and impedes directional transport of photogenerated carriers (Fig. 4e). Therefore, the partial fluorination of TAPT-TPA-2F COFs is more suitable for enhancing electronic coherence and enabling photogenerated electron-hole separation, which promote photocatalytic uranium removal performance.

## Discussion

In summary, this work presents a strategy of leveraging weak electron localization to enhance electronic coherence in COFs, using fluorine atoms as electron-regulating elements to tailor local acceptor states and electronic phase alignment, for greatly boosting photocatalytic uranium removal from nuclear wastewater. This approach enables directional electron transfer for effectively suppressing photogenerated electron–hole recombination and enhancing the SCC efficiency for photocatalytic uranium removal from nuclear wastewater. The optimized TAPT-TPA-2F COFs exhibit a record-high SCC efficiency of 1.52% and nearly 100% uranium removal within the pH range of 3–6, outperforming all previously reported photocatalysts. When applied in a flow-type reactor, the TAPT-TPA-2F COFs demonstrates excellent cycling stability and achieves 99% removal efficiency under natural sunlight, meeting WHO discharge standards and highlighting its practical potential for scalable nuclear wastewater remediation. Both experimental and DFT calculation collectively confirm that electronic coherence enables efficient carrier separation and directional transport for boosting uranium immobilization.

## Methods
### Materials
1,3,5-tris-(4-aminophenyl)triazine (TAPT), n-BuOH, o-dichlorobenzene, acetic acid, terephthalaldehyde (TPA), 2,5-difluoroterephthalaldehyde (TPA-2F) and 2,3,5,6-tetrafluoroterephthalaldehyde (TPA-4F) were purchased from Energy Chemical Co., Ltd, Shanghai Bide Pharmaceutical Technology Co., Ltd.,

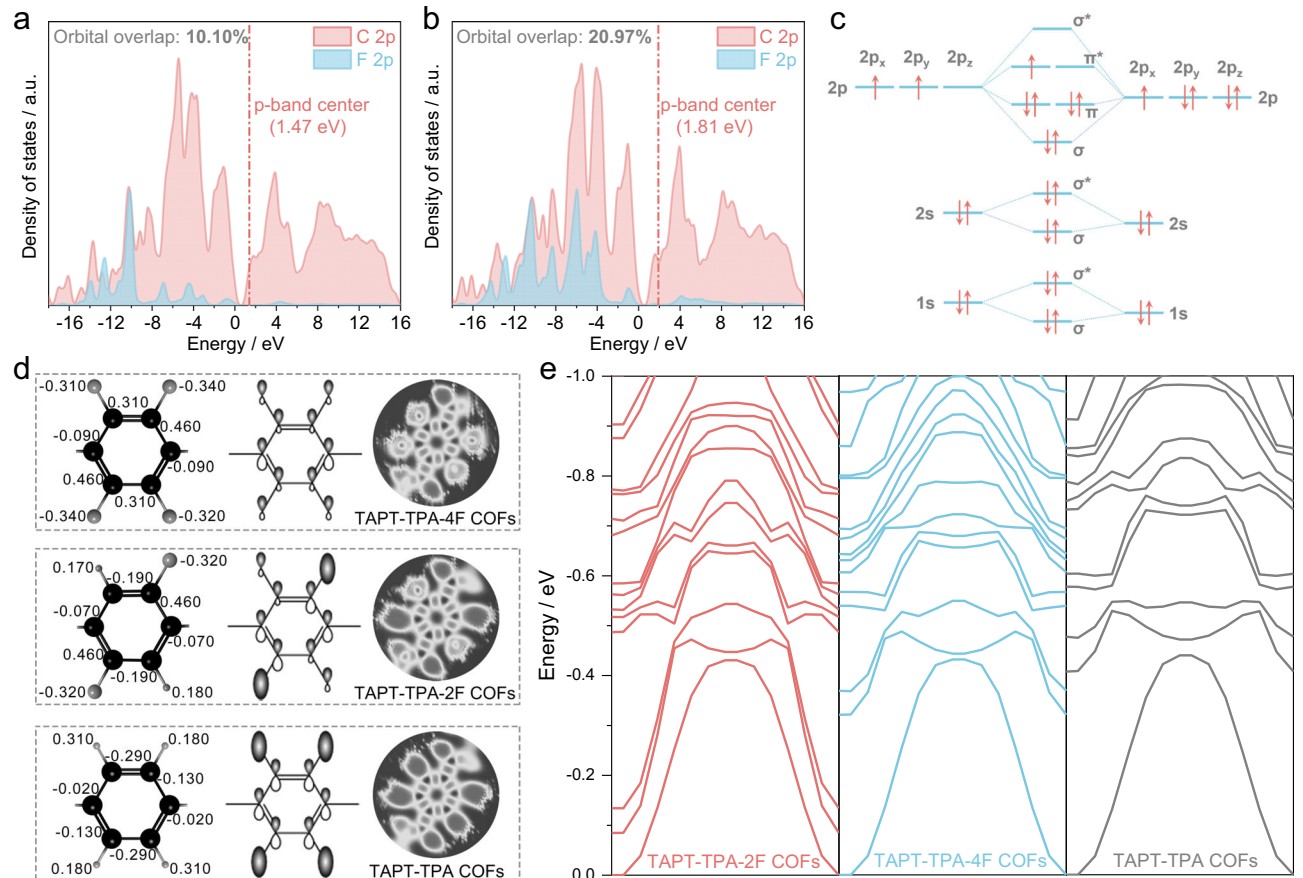

**Fig. 4 | The mechanism of quantum interference enhancing photocatalytic uranium removal. a, b** DOS of C and F atom in TAPT-TPA-2F COFs and TAPT-TPA-4F COFs. **c** Schematic diagram of molecular orbital energy levels and electron arrangement of C-F bond. **d** The Bader charge distribution, orbital hybridization and electron local function of TAPT-TPA-2F COFs, TAPT-TPA-4F COFs and TAPT-TPA COFs. **e** Band distribution of TAPT-TPA-2F COFs, TAPT-TPA-4F COFs and TAPT-TPA COFs.

## Synthesis of TAPT-TPA COFs

The TAPT-TPA COFs were synthesized by acetic acid catalyzed Schiff base reaction of TAPT and TPA organic precursor. Typically, 35 mg of TAPT organic precursor and 22 mg of TPA organic precursor were added into the Pyrex tube. Then, 3 mL mixture of o-dichlorobenzene/n-BuOH (Volume ratio=1:1) was added into the above Pyrex tube, followed by ultrasound for 5 min. Subsequently, 0.1 mL of acetic acid solution (6 M) was added into the above solution. The solution was frozen in liquid nitrogen and then degassed, followed by sealing the Pyrex tube. And then, the Pyrex tube was transferred into oven at 120 °C for 3 d. Finally, the TAPT-TPA COFs was filtered, washed and dried. The synthesis process of TAPT-TPA-2F COFs and TAPT-TPA-4F COFs is the same as that of TAPT-TPA COFs, except that TPA organic precursor is replaced with TPA-2F organic precursor and TPA-4F organic precursor, respectively.

## Photocatalytic removal of uranium

In a typical photocatalytic experiment, 10 mg of COFs and 5 mL of 0.01 M $UO_2^{2+}$ stock solution were added to a 100 mL single-neck flask and diluted to 50 mL with deionized water. The suspension was transferred into a reactor, and the pH was adjusted using 0.1 M NaOH or HCl. Before irradiation, the reactor was bubbled with $N_2$ in the dark for 120 min to remove dissolved oxygen and reach adsorption–desorption equilibrium. A 300 W Xe lamp (Microsolar 300, Beijing Perfectlight) was used as the light source, with a lamp-to-sample distance of 10 cm, an illumination intensity of 100 W $cm^{-2}$ at the reactor surface, and a cut-off filter ($λ > 420$ nm) applied during irradiation. During the reaction, aliquots were withdrawn at defined intervals and analyzed by ICP-MS to determine the $UO_2^{2+}$ concentration.

## Construction of the flow-type quartz microreactor

The flow-type quartz microreactor channel with a width and length of 5 mm and 50 mm, respectively, was built. The photocatalyst was immobilized into the channel using 5% Nafion, and the device was reinforced to prevent the leakage. Simulated nuclear wastewater was introduced through a peristaltic pump at flow rates of 1 mL $min^{-1}$ and 5 mL $min^{-1}$. The flow-type microreactor enables continuous photocatalytic uranium removal under illumination, with the natural sunlight intensity measured at 650 W $m^{-2}$ and its spectral distribution recorded during operation (Supplementary Figs. 67 and 68).

## The determination of the SCC efficiency

The SCC efficiency was determined using the equation:

$$SCC(\%) = \frac{\Delta G_0 \times n_{H_2O_2/\cdot O_2^-}}{P \times S \times t} \times 100\%$$

Herein, the standard Gibbs free energy change ($\Delta G_0$) of photocatalytic $H_2O_2/\cdot O_2^-$ production is 117.000 kJ $mol^{-1}$ and 31.84 kJ $mol^{-1}$, $n_{H_2O_2/\cdot O_2^-}$ is the photocatalytic $H_2O_2/\cdot O_2^-$ yield (mol), P is the input light radiation power (1000 W $m^{-2}$), S is the light irradiated area (about $1 \times 10^{-4}$ $m^2$) and t is the reaction time (s).

## Structural characterization

Powder X-ray diffraction patterns were collected on a Rigaku D/max2400 diffractometer using Cu Kα radiation (λ = 1.5418 Å) to assess the crystallinity and structural features of the COFs. X-ray photoelectron spectroscopy measurements were performed with a monochromated Al Kα source (200 W; Thermo ESCALAB 250) to analyze the surface chemical states and bonding environments, complemented by solid-state NMR spectroscopy (Bruker AVANCE III 600 M) for framework-level structural information. Sample morphologies were examined by transmission electron microscopy (TEM, Tecnai G2 F20 S-Twin). Time-resolved photoluminescence decay profiles were recorded using a fluorescence lifetime spectrophotometer (FLS1000/FSS). Electron paramagnetic resonance (EPR) spectra were obtained on a Bruker EMXPLUS 10/12 spectrometer. UV–vis diffuse reflectance spectra were measured on a Shimadzu UV-3600 spectrophotometer equipped with a diffuse-reflectance accessory, using $BaSO_4$ as the reflectance standard, to determine the optical absorption characteristics and estimate band gaps.

## DFT calculation

DFT computations were conducted using the Cambridge Sequential Total Energy Package (CASTEP, version 19.1.1). Electron–ion interactions were described with the projector-augmented wave (PAW) approach, and exchange–correlation effects were treated within the generalized gradient approximation (GGA) using the Perdew–Burke–Ernzerhof (PBE) functional. A plane-wave basis set was employed with a kinetic-energy cutoff of 700 eV. To account for on-site Coulomb interactions of localized Zn 3d states, a GGA + U scheme was applied with an effective Hubbard parameter of $U_{eff}$ = 8.5 eV. Self-consistent field (SCF) cycles were converged to $1 \times 10^{-6}$ eV in total energy. Brillouin-zone integrations used a $1 \times 1 \times 1$ (Γ-point) k-point mesh together with Gaussian smearing (σ = 0.05 eV). Geometry optimizations were performed using a damped molecular-dynamics algorithm until the residual forces on all atoms were below 0.01 eV Å⁻¹.

## Molecular dynamic simulation

The simulation system was constructed by placing the COF model in a cubic periodic cell $(8.0 \times 8.0 \times 8.0$ nm), corresponding to an initial density of ~0.0510 g cm⁻³. The structure was pre-equilibrated using a five-cycle annealing protocol. The COF was then solvated in an explicit aqueous environment containing 5000 water molecules and 8 uranyl ions. Charge neutrality was maintained by adding $Na^+$ or $Cl^-$ counterions as needed. To reduce computational cost, the uranyl concentration was set to 0.09 mol L⁻¹. Initial configurations were generated with Packmol. All MD simulations were performed using LAMMPS with the consistent valence force field (CVFF). Partial charges for the COFs building units were derived from restrained electrostatic potential (RESP) fitting based on quantum-chemical calculations in an implicit solvation model at the B3LYP/6-31 G* level (Gaussian 09), with the charge fitting carried out using Multiwfn.

The MD workflow comprised: (i) energy minimization to relieve unfavorable contacts; (ii) a 200 ps NVT run at 300 K with initial velocities assigned from a Maxwell–Boltzmann distribution; (iii) a 2 ns NPT equilibration; and (iv) 10 ns production NPT simulations for independent sampling in each case. A 2.0 fs integration time step was used throughout. Temperature was regulated by a velocity-rescaling thermostat (relaxation time 0.1 ps), and pressure was controlled isotropically using the Berendsen barostat (target pressure 10⁵ Pa, time constant 1.0 ps). Long-range electrostatics were evaluated with the particle–mesh Ewald method under periodic boundary conditions in all three dimensions.

## Data availability

The authors declare that all data supporting the conclusions of this study are available within the paper and its Supplementary Information. Source data are provided with this paper. Any additional raw data are available from the corresponding author upon request. Source data are provided with this paper.

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

## Acknowledgements

This study was financially supported by National Key R&D Program of China (No. 2022YFE0128500), Fundamental and Interdisciplinary Disciplines Breakthrough Plan of the Ministry of Education of China, National Science Fund for Distinguished Young Scholars (No. 52025133), National Natural Science Foundation of China (No. 52261135633, No. 52303363), the Beijing Natural Science Foundation (No. Z220020), CNPC Innovation Found (No. 2021DQ02-1002), China National Post-doctoral Program for Innovative Talents (No. BX20220009) and Project funded by China Postdoctoral Science Foundation (No.2022M720225).

## Author contributions

S.G. conceived the project. Y.X. and Y.L. designed and performed the experiments. X.Y. performed the DFT calculation and discussed with Y.L., Y.X. conducted data analyses with the help of Y.L., Y.W., Z.L., Z.S., L.L., N.Y., P.Y., R.Z., M.L. and S.G., All the authors participated in discussions of the research.

## Competing interests

The authors declare no competing interests.
