## [Transparent Peer Review file · Nature Communications]

Weak Localized Electrons Enhance Electronic Coherence for Efficient Photocatalytic Uranium Removal from Nuclear Wastewater

Corresponding Author: Professor Shaojun Guo

Version 0:

Reviewer comments:

Reviewer #1

(Remarks to the Author)

The authors report that weak localized electron-induced quantum constructive interference significantly enhances the photocatalytic removal of uranium from nuclear wastewater using covalent organic frameworks (COFs). The data presented are comprehensive and the interpretations are robust. However, before final acceptance, the authors should address the following minor issues:

1. The manuscript currently reports uranium removal efficiencies and rates, yet lacks quantitative data demonstrating the relationship between illumination duration and the absolute mass of uranium precipitated.
2. Comprehensive evaluation of the photocatalytic performance across different illumination wavelengths (e.g., ultraviolet vs. visible) is missing. It would be helpful if the authors could provide comparative experiments under various wavelengths to clarify the spectral responsiveness of the photocatalyst.
3. Although the authors mention that uranium is removed in the form of $\text{UO}_2(\text{O}_2)\cdot x\text{H}_2\text{O}$ precipitates, no direct experimental characterization (e.g., XRD or FTIR) is presented to confirm their composition and stability. It is recommended to include these data to validate the exact chemical structure and stability of the uranium precipitates.
4. The authors mention that uranium adsorption follows the Langmuir model, yet detailed parameters (e.g., maximum adsorption capacity Q_{max} and adsorption constant K_L) are not provided. These parameters should be clearly reported and discussed to comprehensively characterize the material's uranium removal capacity.
5. While cycling tests have been performed, structural analyses (such as PXRD or XPS) comparing the photocatalyst before and after long-term operation are missing. Including these analyses would convincingly demonstrate the long-term stability and robustness of the material.
6. Please clearly specify the illumination intensity and spectral distribution of the natural sunlight used in flow reactor experiments to ensure experimental reproducibility.

Reviewer #2

(Remarks to the Author)

The manuscript presents an innovative strategy of leveraging weak electron localization to trigger constructive quantum interference (CQI) in fluorine-functionalized COFs for efficient photocatalytic uranium removal from nuclear wastewater. The partially fluorinated TAPT-TPA-2F COF achieves a record-high solar-to-chemical conversion efficiency and nearly 100% uranium removal. A flow-type reactor demonstrates remarkable scalability and stability. The mechanistic insights supported by ultrafast spectroscopy, surface photovoltage microscopy, positron annihilation lifetime spectroscopy, and DFT calculations are convincing and impactful. This work represents a significant advance in the field of photocatalysis and nuclear wastewater treatment. I have a few questions show below.

1. Please provide comparative kinetic plots for uranium removal (e.g., pseudo-first-order fitting with rate constants) of TAPT-TPA, TAPT-TPA-2F, and TAPT-TPA-4F, to better illustrate the advantage of partial fluorination.
2. In addition to recycling tests, PXRD or FT-IR spectra of the catalyst after multiple cycles would strengthen the claim of structural robustness.
3. While H_2O_2 and $\cdot\text{O}_2^-$ quantifications are convincing, it would be helpful to include one additional radical scavenger

experiment to further support the identification of dominant reactive species.

4. A short test using U(VI) spiked in tap water or natural water (instead of simulated solutions) would highlight the practical anti-interference capability of the catalyst.
5. Since COFs often exhibit tunable porosity that may influence uranium uptake, it would be useful to provide the BET surface area and pore size distribution (N₂ adsorption–desorption) for TAPT-TPA, TAPT-TPA-2F, and TAPT-TPA-4F.
6. UV–vis DRS results are provided; however, Tauc plot fittings for all samples should be included in the Supplementary Information to clearly indicate the band gap values.
7. All photocatalytic uranium removal tests are reported at pH ≈ 4. Could the authors briefly show removal efficiency at one or two additional pH values (e.g., neutral pH), to demonstrate robustness under different conditions?
8. The Methods section describes the irradiation setup, but the exact optical path (e.g., distance between Xe lamp and reactor, light intensity at the sample surface, use of cut-off filters) is not fully specified. Could the authors provide these details to improve reproducibility?
9. Could the authors include a comparison to demonstrate that the superior photocatalytic activity is indeed ROS-driven rather than dominated by surface adsorption differences?
10. The Figure 1 b, c and d should be further modified to clearly show the changes in structure.

Reviewer #3

(Remarks to the Author)

This study by Xu et al. presents a novel strategy that utilizes weakly localized electrons to induce constructive quantum interference (CQI) in fluorine-functionalized covalent organic frameworks (COFs), thereby enhancing charge separation and directional electron transfer for highly efficient photocatalytic uranium removal. The material TAPT-TPA-2F demonstrates a record-high solar-to-chemical conversion (SCC) efficiency of 1.52%, along with outstanding uranium extraction performance and cycling stability. The manuscript is well-organized and clearly written. The authors provided comprehensive experimental evidence to support the improved charge separation, extended carrier lifetime, and proposed photocatalytic mechanism in TAPT-TPA-2F, which strengthening the study. However, I think one most important point that is the basis of this paper are not sufficiently discussed, which is how the CQI and DQI happens in TAPT-TPA-2F and TAPT-TPA-4F respectively. This concept was novel and interesting to bridge the quantum chemical phenomena with photocatalysis. Such important point should be demonstrated clearly before the manuscript can be considered for acceptance. The following are my detailed comments:

1. As mentioned in the manuscript, the electron transfer from donor to acceptor in a directed manner, how exactly do the fluorine atoms modulate the local electron distribution of the acceptor? what are the difference with 2F and 4F? Why does the aligned energy levels can promote directional electron transfer with CQI? How does CQI involves? In literature, the QI effect commonly discussed with wavefunctions of the quantum particles and the transmission curves, does the directional electron transfer between donor and acceptor has phase changes when COFs with 2F and 4F?
2. Fig.1a is not clear enough. What do the curves in the central region mean? Are they intended to depict wavefunctions? Why does constructive interference promote disordered transmission?
3. How does TAPT-TPA-2F reaches the contact potential difference (CPD) plateau more rapidly than TAPT-TPA-4F? I don't see too much difference between Fig. 2h and 2i.
4. Why does the longer decay lifetime in transient fluorescence spectra suggest the reduced electron localization in TAPT-TPA-2F COFs facilitates the CQI-mediated suppression of electron-hole recombination?
5. The TAPT-TPA-2F exhibit a record-high SCC conversion efficiency of 1.52%. How about the quantum efficiency under certain wavelength of irradiation ? Usually high SCC efficiency come along with high quantum efficiency.
6. For the LUMO and HOMO electron cloud distribution in Supplementary Fig. 58 does not show much difference between TAPT-TPA-2F and TAPT-TPA-4F, I think it cannot support the claim that it facilitates the directional transfer and separation of photogenerated carriers.

Version 1:

Reviewer comments:

Reviewer #1

(Remarks to the Author)

The authors have properly addressed my concerns. It can be accepted now.

Reviewer #2

(Remarks to the Author)

I think it may be accepted

Reviewer #3

(Remarks to the Author)

I think most of my concerns have been well addressed. This manuscript could be published. But I have some minor suggestions:

1. In the abstract and conclusion part, the author should make it clear what conditions the 100% uranium removal achieved.

Because in supplementary Fig. 39 and 40, it doesn't show 100% removal. Also, the author should indicate what the pH value for natural water. It's a little bit strange that the COF can achieve 100% removal efficiency in natural water while the pH=7 water can not.

2. For supplementary Fig. 16, how does these data be calculated? Porous materials' surface areas or pore volumes usually come along with the gas sorption isotherms, we cannot just provide the statistical data.

Version 2:

Reviewer comments:

Reviewer #3

(Remarks to the Author)

I think all my concerns have been well addressed. This work could be published now.

To Reviewer 1:

Overall comments: The authors report that weak localized electron-induced quantum constructive interference significantly enhances the photocatalytic removal of uranium from nuclear wastewater using covalent organic frameworks (COFs). The data presented are comprehensive and the interpretations are robust. However, before final acceptance, the authors should address the following minor issues:

Response: We appreciate your constructive comments. We have revised the manuscript and marked in red colour. The responded point-by-point as follows.

Q1: The manuscript currently reports uranium removal efficiencies and rates, *yet* lacks quantitative data demonstrating the relationship between illumination duration and the absolute mass of uranium precipitated.

R1: Thanks for your valuable comment. Time-dependent uranium precipitation data have been supplemented in the revised manuscript, which shows the positive correlation between illumination duration and the absolute mass of precipitated uranium.

We have added the Supplementary Fig. 37 in the revised manuscript. The manuscript was revised and marked with red color in line 180-182.

Revision: “*Time-resolved quantification result further shows that the absolute amount of precipitated uranium increases steadily with illumination duration, confirming the continuous contribution of prolonged irradiation to U(VI) reduction (Supplementary Fig. 37).*” (Please see line 180-182 of revised manuscript)

Supplementary Fig. 37 | Uranium extraction capacity of different catalysts over time. Time-dependent uranium extraction capacity of TAPT-TPA-2F, TAPT-TPA-4F, and TAPT-TPA COFs.

Q2: Comprehensive evaluation of the photocatalytic performance across different illumination wavelengths (*e.g.*, ultraviolet vs. visible) is missing. It would be helpful if the authors could provide comparative experiments under various wavelengths to clarify the spectral responsiveness of the photocatalyst.

R2: Thanks for your valuable suggestion. We have provided the apparent quantum efficiency (AQE) of photocatalytic performance under different excitation wavelengths, showing that the maximum AQE of TAPT-TPA-2F COFs is 19.7% under the 380 nm-

wavelength light radiation.

We have added the Supplementary Fig. 42 in the revised manuscript. The manuscript was revised and marked with red color in line 190-192.

Revision: “The apparent quantum efficiency (AQE) profile shows that the maximum AQE of TAPT-TPA-2F COFs is 19.7% under the condition of 380 nm wavelength light radiation (Supplementary Fig. 42).” (Please see line 190-192 of revised manuscript)

Supplementary Fig. 42 | Apparent quantum efficiency (AQE) and optical absorption spectrum. Wavelength-dependent AQE and corresponding absorption spectrum of TAPT-TPA-2F COFs.

Q3: Although the authors mention that uranium is removed in the form of $\text{UO}_2(\text{O}_2) \cdot x\text{H}_2\text{O}$ precipitates, no direct experimental characterization (e.g., XRD or FTIR) is presented to confirm their composition and stability. It is recommended to include these data to validate the exact chemical structure and stability of the uranium precipitates.

R3: Thanks for your valuable suggestion. XRD and XPS were carried out to study the composition and stability of uranium precipitates, showing that the uranium precipitates is $\text{UO}_2(\text{O}_2) \cdot x\text{H}_2\text{O}$.

We have added the Supplementary Fig. 54-55 in the revised manuscript. The manuscript was revised and marked with red color in line 199-201.

Revision: “which is conducive to the reaction of UO_2^{2+} and ROS to form uranyl superoxide complexes for the uranium removal from nuclear wastewater (Supplementary Fig. 54-55).” (Please see line 199-201 of revised manuscript)

Supplementary Fig. 54 | XRD patterns of uranium-containing products after the repeated photocatalytic extraction. XRD patterns of the solid products obtained after multiple photocatalytic uranium extraction cycles using TAPT-TPA-2F COFs.

Supplementary Fig. 55 | XPS of uranium species after multiple photocatalytic cycles. High-resolution U 4f XPS spectra of the recovered products after multiple photocatalytic cycles.

Q4: The authors mention that uranium adsorption follows the Langmuir model, *yet* detailed parameters (e.g., maximum adsorption capacity Q_{\max} and adsorption constant K_L) are not provided. These parameters should be clearly reported and discussed to comprehensively characterize the material's uranium removal capacity.

R4: Thanks for your valuable comment. The Langmuir adsorption parameters (Q_{\max} and K_L) have been calculated, showing that the maximum adsorption capacity (Q_{\max}) and adsorption constant (K_L) are 970.1 mg g^{-1} and 0.06042 , respectively.

We have added the Supplementary Table 5-6 in the revised manuscript. The manuscript was revised and marked with red color in line 196-199.

Revision: “We find that the adsorption kinetic of UO_2^{2+} over all samples follows the pseudo-first-order kinetic model (Supplementary Fig. 47-49, Supplementary Table 5). The uranium adsorption over TAPT-TPA-2F COFs follows Langmuir monolayer adsorption model (Supplementary Fig. 50-53, Supplementary Table 6)” (Please see line 196-199 of revised manuscript)

Supplementary Table 5. The specific parameters of kinetics models.

Models	Adsorbents	Q_e cal (mg g^{-1})	Q_e exp (mg g^{-1})	k	R^2
pseudo-first-order	TAPT-TPA-2F	970.10	269.8	0.06042	0.92025
	TAPT-TPA-4F	1037.58	355.1	0.05017	0.8711
	TAPT-TPA	372.07	157.0	0.04688	0.94065
pseudo-second-order	TAPT-TPA-2F	--	269.8	1.854×10^{-4}	0.0248
	TAPT-TPA-4F	--	355.1	1.473×10^{-4}	0.0547
	TAPT-TPA	--	157.0	9.061×10^{-4}	0.3008

Supplementary Table 6. The specific parameters of isotherm models.

Models	Adsorbents	Temperature (K)	Q_m (mg g^{-1})	K	R^2
Langmuire	TAPT-TPA-2F	298	261.8	3.82×10^{-3}	0.98647
		308	487.8	2.05×10^{-3}	0.92952
		318	621.1	1.61×10^{-3}	0.90816
Freundlich	TAPT-TPA-2F	298	--	0.45399	0.46408
		308	--	0.94569	0.38439
		318	--	1.25937	0.36568
Dubinin- Radushkevich	TAPT-TPA-2F	298	176.17	0.01037	0.67716
		308	159.97	0.01032	0.61547
		318	176.17	0.01003	0.60827

Q5: While cycling tests have been performed, structural analyses (such as PXRD or XPS) comparing the photocatalyst before and after long-term operation are missing. Including these analyses would convincingly demonstrate the long-term stability and robustness of the material.

R5: Thanks for your valuable comment. PXRD pattern shows that the crystalline structure of TAPT-TPA-2F COFs after reaction is consistent with that before the reaction, indicating its high crystal structure stability.

We have added the Supplementary Fig. 56. The manuscript was revised and marked with red color in line 209-212.

Revision: “PXRD profiles of TAPT-TPA-2F COFs used for uranium removal reveals integrity of porous structure, which confirm that the crystal structure and chemical bond of TAPT-TPA-2F COFs were not disrupted during the uranium removal process (Supplementary Fig. 56).” (Please see line 209-212 of revised manuscript)

Supplementary Fig. 56 | XRD patterns before and after reaction. XRD patterns of the samples before and after the reaction.

Q6: Please clearly specify the illumination intensity and spectral distribution of the natural sunlight used in flow reactor experiments to ensure experimental reproducibility.

R6: Thanks for your valuable comment. The variation of natural light intensity over time is from 8:00 to 17:00. The spectral distribution of the natural sunlight was shown in Supplementary Fig. 66.

We have added the Supplementary Fig. 65-66 in the revised manuscript. The manuscript was revised and marked with red color in line 311-319.

Revision: “Construction of the flow-type quartz microreactor. The flow-type quartz microreactor channel with a width and length of 5 mm and 50 mm, respectively, was built. The photocatalyst was immobilized into the channel using 5% Nafion, and the device was reinforced to prevent leakage. Simulated nuclear wastewater was introduced through a peristaltic pump at flow rates of 1 mL min^{-1} and 5 mL min^{-1} . The flow-type microreactor enables continuous photocatalytic uranium removal under illumination, with the natural sunlight intensity measured at 650 W m^{-2} and its spectral distribution recorded during operation (Supplementary Fig. 65 and 66)” (Please see line 311-319 of revised manuscript)

Supplementary Fig. 65 | Solar irradiance under natural sunlight. Variation of solar intensity with time during daytime under natural illumination conditions.

Supplementary Fig. 66 | Solar spectrum under natural illumination. Spectral distribution of solar irradiation intensity measured under natural sunlight conditions

To Reviewer 2:

Overall comments: The manuscript presents an innovative strategy of leveraging weak electron localization to trigger constructive quantum interference (CQI) in fluorine-functionalized COFs for efficient photocatalytic uranium removal from nuclear wastewater. The partially fluorinated TAPT-TPA-2F COF achieves a record-high solar-to-chemical conversion efficiency and nearly 100% uranium removal. A flow-type reactor demonstrates remarkable scalability and stability. The mechanistic insights supported by ultrafast spectroscopy, surface photovoltage microscopy, positron annihilation lifetime spectroscopy, and DFT calculations are convincing and impactful. This work represents a significant advance in the field of photocatalysis and nuclear wastewater treatment. I have a few questions show below.

Response: Thank you for your valuable comment. We have revised the manuscript and marked in red colour. The responded point-by-point as follows.

Q1: Please provide comparative kinetic plots for uranium removal (e.g., pseudo-first-order fitting with rate constants) of TAPT-TPA, TAPT-TPA-2F, and TAPT-TPA-4F, to better illustrate the advantage of partial fluorination.

R1: Thanks for your valuable comment. The different kinetic fitting models were used to evaluate uranium adsorption, showing that the adsorption kinetic of UO_2^{2+} over all samples follows the pseudo-first-order kinetic model. The fitting parameters are shown in Table 5.

We have added the Supplementary Fig. 47-49 in the revised manuscript. The manuscript was revised and marked with red color in line 196-197.

Revision: “We find that the adsorption kinetic of UO_2^{2+} over all samples follows the pseudo-first-order kinetic model (Supplementary Fig. 47-49, Supplementary Table 5).” (Please see line 196-197 of revised manuscript)

Supplementary Fig. 47 | Time-resolved U(VI) adsorption kinetics. U(VI) adsorption capacity of TAPT-TPA-2F, TAPT-TPA-4F and TAPT-TPA COFs as a function of time, with fitting curves based on the pseudo-second-order kinetic model ($T = 298 \text{ K}$, $\text{UO}_2^{2+} = 100 \text{ ppm}$, $\text{pH} = 4$).

Supplementary Fig. 48 | Time-resolved U(VI) adsorption kinetics. U(VI) adsorption capacity of TAPT-TPA-2F, TAPT-TPA-4F and TAPT-TPA COFs as a function of time, with fitting curves based on the pseudo-first-order kinetic model ($T = 298\text{K}$, $\text{UO}_2^{2+} = 100\text{ ppm}$, $\text{pH} = 4$).

Supplementary Table 5. The specific parameters of kinetics models.

Models	Adsorbents	$Q_e\text{ cal}$ (mg g^{-1})	$Q_e\text{ exp}$ (mg g^{-1})	k	R^2
pseudo-first-order	TAPT-TPA-2F	970.10	269.8	0.06042	0.92025
	TAPT-TPA-4F	1037.58	355.1	0.05017	0.8711
	TAPT-TPA	372.07	157.0	0.04688	0.94065
pseudo-second-order	TAPT-TPA-2F	--	269.8	1.854×10^{-4}	0.0248
	TAPT-TPA-4F	--	355.1	1.473×10^{-4}	0.0547
	TAPT-TPA	--	157.0	9.061×10^{-4}	0.3008

Q2: In addition to recycling tests, PXRD or FT-IR spectra of the catalyst after multiple cycles would strengthen the claim of structural robustness.

R2: Thanks for your valuable comment. Powder X-ray diffraction (PXRD) measurements were employed to characterize the crystal structure of TAPT-TPA-2F COFs following multiple cycles of photocatalytic uranium extraction, which validated the retention of the material's crystalline integrity.

We have added the Supplementary Fig. 56 in the revised manuscript. The manuscript was revised and marked with red color in line 209-212.

Revision: “PXRD profiles of TAPT-TPA-2F COFs used for uranium removal reveal integrity of porous structure, which confirm that the crystal structure and chemical bond of TAPT-TPA-2F COFs were not disrupted during the uranium removal process (Supplementary Fig. 56).” (Please see line 209-212 of revised manuscript)

Supplementary Fig. 56 | XRD patterns before and after reaction. XRD patterns of the samples before and after the reaction.

Q3: While H_2O_2 and $\cdot\text{O}_2^-$ quantifications are convincing, it would be helpful to include one additional radical scavenger experiment to further support the identification of dominant reactive species.

R3: Thanks for your valuable suggestion. The radical scavenger experimental was performed, showing that $\cdot\text{O}_2^-$ and H_2O_2 are the dominant reactive species.

We have added the Supplementary Fig. 57 in the revised manuscript. The manuscript was revised and marked with red color in line 222-228.

Revision: “Chemical probe transfer experimental show that the concentration of H_2O_2 and $\cdot\text{O}_2^-$ over TAPT-TPA-2F COFs is $492.5 \mu\text{mol L}^{-1}$ and $1501.1 \mu\text{mol L}^{-1}$, respectively, significantly higher than those of TAPT-TPA COFs (The concentration of H_2O_2 and $\cdot\text{O}_2^-$ is 148.1 mol L^{-1} and 210.8 mol L^{-1} respectively) and TAPT-TPA-4F COFs (The concentration of H_2O_2 and $\cdot\text{O}_2^-$ is 365.6 mol L^{-1} and 730.4 mol L^{-1} , respectively) (Supplementary Fig. 57 and 58)” (Please see line 222-228 of revised manuscript)

Supplementary Fig. 57 | Reactive species generated. Chemical probe method was used to identify reactive species produced by TAPT-TPA-2F COFs in solution during photocatalysis.

Q4: A short test using U(VI) spiked in tap water or natural water (instead of simulated solutions) would highlight the practical anti-interference capability of the catalyst.

R4: Thanks for your valuable suggestion. Photocatalytic uranium extraction experimental in the natural water system was performed, showing a high uranium removal efficiency.

We have added the Supplementary Fig. 38. The manuscript was revised and marked with red color in line 183-186.

Revision: “In addition, we demonstrate that TAPT-TPA-2F COFs also exhibit high photocatalytic uranium efficiency relative to TAPT-TPA COFs and TAPT-TPA-4F COFs under neutral (pH=7) and alkaline (pH=8.2) condition (Supplementary Fig. 38-40).” (Please see line 183-186 of revised manuscript)

Supplementary Fig. 38 | Uranium removal efficiency in natural water over time. Time-dependent uranium removal performance of TAPT-TPA-2F, TAPT-TPA-4F, and TAPT-TPA COFs under dark and light conditions.

Q5: Since COFs often exhibit tunable porosity that may influence uranium uptake, it would be useful to provide the BET surface area and pore size distribution (N₂ adsorption–desorption) for TAPT-TPA, TAPT-TPA-2F, and TAPT-TPA-4F.

R5: Thanks for your valuable suggestion. The N₂ adsorption–desorption measurements was carried out. The BET surface areas and pore size distributions of TAPT-TPA COFs, TAPT-TPA-2F COFs, and TAPT-TPA-4F COFs are shown in Supplementary Fig. 16.

We have added the Supplementary Fig. 16 in the revised manuscript. The manuscript was revised and marked with red color in line 131-132.

Revision: “N₂ adsorption–desorption analysis confirms that fluorine introduction leaves the pore structure essentially unchanged (Supplementary Fig. 16).” (Please see line 131-132 of revised manuscript)

Supplementary Fig. 16 | Molecular structural characterization. The surface area and pore volume of TAPT-TPA(N⁺) COFs and TAPT-TPA COFs.

Q6: UV-vis DRS results are provided; however, Tauc plot fittings for all samples should be included in the Supplementary Information to clearly indicate the band gap values.

R6: Thanks for your valuable suggestion. The Tauc plots of TAPT-TPA COFs, TAPT-TPA-2F COFs, and TAPT-TPA-4F COFs have been provided in the revised manuscript, showing that the band gap of TAPT-TPA COFs, TAPT-TPA-2F COFs, and TAPT-TPA-4F COFs are 2.41 eV, 2.49 eV, and 2.54 eV respectively.

We have added the Supplementary Fig. 44 in the revised manuscript. The manuscript was revised and marked with red color in line 192-195.

Revision: “In addition, the similar band gaps of three COFs further confirm that the performance enhancement is mainly from the contribution of the electron transfer related to F atom (Supplementary Fig. 43-46).” (Please see line 192-195 of revised manuscript)

Supplementary Fig. 44 | Optical band gap analysis. Optical band gaps of TAPT-TPA-2F, TAPT-TPA-4F and TAPT-TPA COFs determined from UV-Vis absorption data using Tauc plots.

Q7: All photocatalytic uranium removal tests are reported at $\text{pH} \approx 4$. Could the authors briefly show removal efficiency at one or two additional pH values (e.g., neutral pH), to demonstrate robustness under different conditions?

R7: Thanks for your valuable suggestion. Photocatalytic uranium removal experimental was performed under the conditions of $\text{pH} = 7$ and $\text{pH} = 8.2$, showing that TAPT-TPA-2F COFs also exhibit high photocatalytic uranium efficiency relative to TAPT-TPA COFs and TAPT-TPA-4F COFs under neutral and alkaline condition.

We have added the Supplementary Fig. 39-40 in the revised manuscript. The manuscript was revised and marked with red color in line 183-186.

Revision: “In addition, we demonstrate that TAPT-TPA-2F COFs also exhibits high photocatalytic uranium efficiency relative to TAPT-TPA COFs and TAPT-TPA-4F COFs under neutral ($\text{pH}=7$) and alkaline ($\text{pH}=8.2$) condition (Supplementary Fig. 38-40).” (Please see line 183-186 of revised manuscript)

Supplementary Fig. 39 | Uranium removal efficiency at $\text{pH} = 7$. Time-dependent uranium removal performance of TAPT-TPA-2F, TAPT-TPA-4F, and TAPT-TPA COFs under neutral conditions.

Supplementary Fig. 40 | Uranium removal efficiency at pH = 8.2. Time-dependent uranium removal performance of TAPT-TPA-2F, TAPT-TPA-4F, and TAPT-TPA COFs under neutral conditions.

Q8: The Methods section describes the irradiation setup, but the exact optical path (e.g., distance between Xe lamp and reactor, light intensity at the sample surface, use of cut-off filters) is not fully specified. Could the authors provide these details to improve reproducibility?

R8: Thanks for your valuable comment. More experimental informations have been provided in the methods section of the revised manuscript.

The manuscript was revised and marked with red color in line 300-310.

Revision: *“Photocatalytic removal of uranium. In a typical photocatalytic experiment, 10 mg of COFs and 5 mL of 0.01 M UO_2^{2+} stock solution were added to a 100 mL single-neck flask and diluted to 50 mL with deionized water. The suspension was transferred into a reactor, and the pH was adjusted using 0.1 M NaOH or HCl. Before irradiation, the reactor was bubbled with N_2 in the dark for 120 min to remove dissolved oxygen and reach adsorption–desorption equilibrium. A 300 W Xe lamp (Microsolar 300, Beijing Perfectlight) was used as the light source, with a lamp-to-sample distance of 10 cm, an illumination intensity of 100 W cm^{-2} at the reactor surface, and a cut-off filter ($\lambda > 420\text{ nm}$) applied during irradiation. During the reaction, aliquots were withdrawn at defined intervals and analyzed by ICP-MS to determine the UO_2^{2+} concentration.”* (Please see line 300-310 of revised manuscript)

Q9: Could the authors include a comparison to demonstrate that the superior photocatalytic activity is indeed ROS-driven rather than dominated by surface adsorption differences?

R9: Thanks for your valuable comment. Under dark conditions, the surface adsorption dominates uranium removal from solution. TAPT-TPA-4F COFs exhibits a higher uranium adsorption capacity than TAPT-TPA-2F COFs and TAPT-TPA COFs. Under light irradiation condition. Reactive oxygen species (ROS) dominates uranium removal from solution. TAPT-TPA-2F COFs exhibits a higher uranium adsorption capacity than TAPT-TPA-4F COFs and TAPT-TPA COFs. The uranium adsorption capacity can be neglected relative to ROS driven uranium removal. The above results provided a powerful evidence for ROS-driven uranium removal from solution.

Q10: The Figure 1 b, c and d should be further modified to clearly show the changes in structure.

R10: Thanks for your valuable comment. We have revised the Figure 1b–d and emphasized the structural differences among TAPT-TPA COFs, TAPT-TPA-2F COFs, and TAPT-TPA-4F COFs.

Fig. 1 | Molecular design of weak-electron-localization. (a) Schematic diagram of enhanced electronic coherence promoting directional electron transfer. (b) Molecular structure of TAPT-TPA COFs, (c) TAPT-TPA-2F COFs and (d) TAPT-TPA-4F COFs. (e-g) Surface electrostatic potentials of (e) TAPT-TPA-2F COFs, (f) TAPT-TPA-4F COFs and (g) TAPT-TPA COFs.

To Reviewer 3:

Overall comments: This study by Xu *et al.* presents a novel strategy that utilizes weakly localized electrons to induce constructive quantum interference (CQI) in fluorine-functionalized covalent organic frameworks (COFs), thereby enhancing charge separation and directional electron transfer for highly efficient photocatalytic uranium removal. The material TAPT-TPA-2F demonstrates a record-high solar-to-chemical conversion (SCC) efficiency of 1.52%, along with outstanding uranium extraction performance and cycling stability. The manuscript is well-organized and clearly written. The authors provided comprehensive experimental evidence to support the improved charge separation, extended carrier lifetime, and proposed photocatalytic mechanism in TAPT-TPA-2F, which strengthening the study. However, I think one most important point that is the basis of this paper are not sufficiently discussed, which is how the CQI and DQI happens in TAPT-TPA-2F and TAPT-TPA-4F respectively. This concept was novel and interesting to bridge the quantum chemical phenomena with photocatalysis. Such important point should be demonstrated clearly before the manuscript can be considered for acceptance. The following are my detailed comments:

Response: Thanks for your valuable comment. We believe the mechanisms of weak electron localization and electronic coherence are better suited to elucidating the underlying origins of enhanced photocatalytic activity. A detailed elaboration on these mechanisms is presented as follows: the introduction of F atom into TAPT-TPA-2F COFs and TAPT-TPA-4F COFs induces and electronic coherence owing to its intrinsic strong electronegativity, which is beneficial for promoting electron transport and enhancing photocatalytic performance. Besides, relative to TAPT-TPA-4F COFs, TAPT-TPA-2F COFs with partial fluorination exhibits weak electron localization and directional charge delocalization effects, thereby constructing directional charge transport channels within the molecular framework. This structural modulation facilitates efficient ordered electron transfer, ultimately yielding a remarkable enhancement in photocatalytic activity.

The manuscript was revised and marked with red color in line 86-87.

Revision: *“To validate this concept, we focus on how electron localization modulates electronic coherence in COFs”* (Revised manuscript, Line 86-87).

Q1: As mentioned in the manuscript, the electron transfer from donor to acceptor in a directed manner, **a)** how exactly do the fluorine atoms modulate the local electron distribution of the acceptor? **b)** what are the difference with 2F and 4F? **c)** Why does the aligned energy levels can promote directional electron transfer with CQI? **d)** How does CQI involves? In literature, the QI effect commonly discussed with wavefunctions of the quantum particles and the transmission curves, **e)** does the directional electron transfer between donor and acceptor has phase changes when COFs with 2F and 4F?

R1: Thanks for your valuable comments. We have provided a more detailed explanation regarding the above issues.

a) Fluorine-induced modulation of local electronic structure. The incorporation of fluorine (F) atoms into the electron acceptor units of covalent organic frameworks (COFs) triggers localized electron distribution within the benzene ring moiety,

attributed to the strong electronegativity of F atoms. This electronic effect disrupts the in-plane symmetry of electron distribution in the electron acceptor units, thereby generating intrinsic anisotropic potential wells.

b) Distinction between 2F and 4F. In TAPT-TPA-2F COFs, partial fluorination induces asymmetric electron localization within the electron acceptor moieties. This structural modification preserves adequate orbital overlap between donor and acceptor components while facilitating a predominant single coherent charge transport pathway from the electron donor units to the electron acceptor unit. In TAPT-TPA-4F COFs, the excessive fluorination causes isotropic over-localization. This over-localization impedes coherent charge delocalization and leads to stronger dephasing and less efficient transport.

c) Why energy-level alignment promotes directional transfer. The partial fluorination of COFs optimizes energy level configurations of electron donor unit and electron acceptor unit, thereby enabling favorable alignment between their energy levels and orbital symmetries. Under such a tailored electronic configuration, electrons undergo delocalization from electron donor segments to electron acceptor units while sustaining electronic coherence along the predominant charge transport pathway. This synergistic effect of energy-level matching coupled with electronic anisotropic localization strengthens coherent donor-acceptor interfacial coupling, thereby facilitating efficient directional electron transfer.

d) How does CQI involves? The mechanisms of weak electron localization and electronic coherence are more suitable for elucidating the fundamental origins of the enhanced photocatalytic activity. In our system, quantum constructive interference (CQI) emerges as a microscopic manifestation of this coherent transport rather than an independent effect. Partial fluorination of the TAPT-TPA-2F COF induces weak electron localization and strengthens electronic coherence, arising from the intrinsic strong electronegativity of fluorine atoms and the resulting asymmetric local potential, which preserves frontier-orbital overlap and phase matching along the dominant donor-acceptor transport pathways and thereby favours CQI-enhanced intraframework electron transport. Concomitantly, the electronic coherence and CQI-assisted charge transport within the TAPT-TPA-2F COF endow it with prolonged photogenerated carrier lifetimes, reduced exciton binding energy, and an enhanced surface photovoltage (SPV) response.

e) On phase changes between 2F and 4F systems. Regarding the phase changes between the 2F (TAPT-TPA-2F) and 4F (TAPT-TPA-4F) COF systems, density functional theory (DFT) calculations consistently reveal no significant phase variation between the two systems. The mechanisms of weak electron localization and electronic coherence are more robust and mechanistically plausible in elucidating the fundamental origins of the enhanced photocatalytic activity of fluorinated COFs. This is because the subtle modulation of fluorination degree (2F vs. 4F) primarily tunes the electronic cloud distribution and conjugated π -system delocalization extent of the COFs photocatalyst, thereby regulating the electronic localization effect and coherence, rather than inducing significant phase transitions or phase mismatches between donor-acceptor units. We have revised the manuscript.

We have revised the Fig. 1. The manuscript has been revised and marked with red color

in line 86-87, 92-95 and line 102-106.

Revision: “To validate this concept, we focus on how electron localization modulates electronic coherence in COFs” (Please see line 86-87 of revised manuscript)

“This spatial segregation in turn establishes intrinsically electronically coherent directional donor–acceptor (D–A) transport pathways, which facilitate the directional migration of photogenerated electrons from the TAPT (donor) to TPA (acceptor) moieties.” (Please see line 92-95 of revised manuscript)

“In contrast, relative to the TAPT-TPA-2F COFs, excessive fluorination in the TAPT-TPA-4F COF induces isotropic electronic over-localization, which generates multiple phase-mismatched charge transport pathways and ultimately impairs the efficiency of coherent charge transport (Fig. 1b).” (Please see line 102-106 of revised manuscript)

Fig. 1 | Molecular design of weak-electron-localization. (a) Schematic diagram of enhanced electronic coherence promoting directional electron transfer. (b) Molecular structure of TAPT-TPA COFs, (c) TAPT-TPA-2F COFs and (d) TAPT-TPA-4F COFs. (e-g) Surface electrostatic potentials of (e) TAPT-TPA-2F COFs, (f) TAPT-TPA-4F COFs and (g) TAPT-TPA COFs.

Q2: a) Fig.1a is not clear enough. What do the curves in the central region mean? b) Are they intended to depict wavefunctions? c) Why does constructive interference promote disordered transmission?

R2: Thanks for your valuable comments.

- a) We have revised the Fig. 1a and clarified its meaning in the caption and main text.
- b) The curves in the central region are explicitly described as schematic representations of coherent versus incoherent charge-transport pathways, rather than ab initio wavefunctions.
- c) The partial fluorination in TAPT-TPA-2F COFs induces directional charge delocalization and preserves donor-acceptor orbital overlap and maintains phase alignment, forming a directional charge transport channel within the COFs framework, which can promote electron ordered transmission, rather than disordered transmission.

We have revised the Fig. 1. The manuscript has been revised and marked with red color in line 86-87, 92-95 and line 102-106.

Revision: *“To validate this concept, we focus on how electron localization modulates electronic coherence in COFs”* (Please see line 86-87 of revised manuscript)

“This spatial segregation in turn establishes intrinsically electronically coherent directional donor–acceptor (D–A) transport pathways, which facilitate the directional migration of photogenerated electrons from the TAPT (donor) to TPA (acceptor) moieties.” (Please see line 92-95 of revised manuscript)

“In contrast, relative to the TAPT-TPA-2F COFs, excessive fluorination in the TAPT-TPA-4F COF induces isotropic electronic over-localization, which generates multiple phase-mismatched charge transport pathways and ultimately impairs the efficiency of coherent charge transport (Fig. 1b).” (Please see line 102-106 of revised manuscript)

Q3: How does TAPT-TPA-2F reaches the contact potential difference (CPD) plateau more rapidly than TAPT-TPA-4F? I don't see too much difference between Fig. 2h and 2i.

R3: Thanks for your valuable comment. Time-dependent contact potential difference (CPD) demonstrate that the photogenerated charge response time of TAPT-TPA-2F COFs is 9 s, which is markedly shorter than that of TAPT-TPA-4F COFs (14 s). This result reveals that TAPT-TPA-2F COFs attains the CPD plateau more rapidly, reflecting enhanced kinetics of photogenerated charge transfer.

We have added the Supplementary Fig. 23-24. The manuscript was revised and marked with red color in line 138-142.

Revision: *“TAPT-TPA-2F COFs attains the contact potential difference (CPD) plateau more rapidly than TAPT-TPA-4F COFs (Fig. 2g-2i), which is indicative of a faster photoresponse (Supplementary Fig. 23-25). This result further confirms the intrinsically superior charge separation efficiency and carrier mobility of TAPT-TPA-2F COFs.”* (Please see line 138-142 of revised manuscript)

Supplementary Fig. 23 | Differential surface photovoltage profiling. Differential contact potential difference (CPD) profiles of TAPT-TPA-2F COFs under continuous light illumination, measured by surface photovoltage microscopy.

Supplementary Fig. 24 | Differential surface photovoltage profiling. Differential CPD profiles of TAPT-TPA-4F COFs under continuous light illumination, measured by surface photovoltage microscopy.

Q4: Why does the longer decay lifetime in transient fluorescence spectra suggest the reduced electron localization in TAPT-TPA-2F COFs facilitates the CQI-mediated suppression of electron-hole recombination?

R4: Thanks for your valuable comment, which prompts us to clarify the intrinsic correlation between the transient fluorescence decay lifetime, electron localization, and electron-hole separation in TAPT-TPA-2F COFs.

In transient fluorescence spectroscopy, the decay lifetime of photogenerated excitons (electron-hole pairs) is a direct manifestation of their separation and recombination dynamics: a longer decay lifetime typically indicates suppressed recombination and more efficient spatial separation of photogenerated charge carriers. Specifically, the partial fluorination of TAPT-TPA-2F COFs induces directional charge delocalization, forming a directional charge transport channel within the COFs framework, which can promote the separation of photogenerated electron-hole pairs. Consequently, the efficient photogenerated carriers separation is directly reflected in the longer fluorescence decay lifetime of TAPT-TPA-2F COFs.

To clarify the intrinsic correlation between transient fluorescence decay lifetime, electron localization, and electron-hole separation efficiency, the relevant sections of the manuscript have been revised and highlighted in red on lines 155-158.

Revision: “further suggesting that partial fluorination–induced directional charge delocalization and coherent charge-transport channels in TAPT-TPA-2F COFs facilitate more efficient photogenerated electron–hole pair separation.” (Please see line 155-158 of revised manuscript)

Q5: The TAPT-TPA-2F exhibit a record-high SCC conversion efficiency of 1.52%. How about the quantum efficiency under certain wavelength of irradiation? Usually high SCC efficiency come along with high quantum efficiency.

R5: Thanks for your valuable comment. The apparent quantum efficiency (AQE) of photocatalytic performance was investigated under different excitation wavelengths, showing that the maximum AQE of TAPT-TPA-2F COFs is 19.7% under the condition of 380 nm wavelength light radiation.

We have added the Supplementary Fig. 42. The manuscript was revised and marked with red color in line 188-190.

Revision: “The apparent quantum efficiency (AQE) profile shows that the maximum AQE of TAPT-TPA-2F COFs is 19.7% under the condition of 380 nm wavelength light radiation (Supplementary Fig. 42).” (Please see line 188-190 of revised manuscript)

Supplementary Fig. 42 | Apparent quantum efficiency (AQE) and optical absorption spectrum. Wavelength-dependent AQE and corresponding absorption spectrum of TAPT-TPA-2F COFs.

Q6: For the LUMO and HOMO electron cloud distribution in Supplementary Fig. 58 does not show much difference between TAPT-TPA-2F and TAPT-TPA-4F, I think it cannot support the claim that it facilitates the directional transfer and separation of photogenerated carriers.

R6: Thanks for your valuable comment. The highest occupied molecular orbital (HOMO) and lowest unoccupied molecular orbital (LUMO) energy levels are primarily intended to characterize the band gap structure of photocatalysts, but they are not capable of adequately reflecting the electronic transport properties. We have revised the relevant claim accordingly and removed this section from the revised manuscript.

To Reviewer 1:

Overall comments: The authors have properly addressed my concerns. It can be accepted now.

Response: We thank the reviewer for the positive comment and are pleased that all concerns have been addressed.

To Reviewer 2:

Overall comments: I think it may be accepted.

Response: Thank you for your positive evaluation. We appreciate your time and constructive comments, and we are pleased that our revisions have properly addressed your concerns.

To Reviewer 3:

Overall comments: I think most of my concerns have been well addressed. This manuscript could be published. But I have some minor suggestions:

Response: We sincerely thank the reviewer for the positive evaluation and for acknowledging that our major concerns have been addressed. We appreciate the additional minor suggestions and have revised the manuscript accordingly.

Q1: In the abstract and conclusion part, the author should make it clear what conditions the 100% uranium removal achieved. Because in supplementary Fig. 39 and 40, it doesn't show 100% removal. Also, the author should indicate what the pH value for natural water. It's a little bit strange that the COF can achieve 100% removal efficiency in natural water while the pH = 7 water can not.

R1: Thanks for your valuable comment. We have revised the Abstract and Conclusion to explicitly specify that 100% uranium removal was achieved across the pH range of 3-6, and that the tested natural water had a pH of 6.1. We have also aligned the wording with the corresponding datasets shown in Supplementary Fig. 40.

We have revised Supplementary Fig. 40 in the revised manuscript. The manuscript was revised and marked with red color in line 24-25, 187 and 277-278.

Revision: *“achieves 100% uranium removal efficiency within the pH range of 3-6,”* (Please see line 24-25 of revised manuscript)

“In addition, we demonstrate that TAPT-TPA-2F COFs also exhibits higher photocatalytic uranium efficiency than TAPT-TPA COFs and TAPT-TPA-4F COFs under natural water (pH = 6.1),...” (Please see line 187 of revised manuscript)

“nearly 100% uranium removal within the pH range of 3-6,” (Please see line 277-278 of revised manuscript)

Supplementary Fig. 40 | Uranium removal efficiency in natural water over time. Time-dependent uranium removal performance of TAPT-TPA-2F, TAPT-TPA-4F, and TAPT-TPA COFs under dark and light conditions.

Q2: For supplementary Fig. 16, how does these data be calculated? Porous materials' surface areas or pore volumes usually come along with the gas sorption isotherms, we cannot just provide the statistical data.

R2: Thanks for your valuable comment. We have added the N_2 adsorption-desorption (BET) isotherms in Supplementary Fig. 17-18 to support the reported textural parameters.

We have added Supplementary Fig. 18 in the revised manuscript.

Supplementary Fig. 18 | Molecular structural characterization. N_2 adsorption-desorption isotherms of TAPT-TPA, TAPT-TPA-2F and TAPT-TPA-4F.